# Prevalence and Severity of Periodontitis in Patients with Ulcerative Colitis: A Case–Control Study

**DOI:** 10.3390/ijerph22091355

**Published:** 2025-08-29

**Authors:** Angar Soronzonbold, Oyunkhishig Khishigdorj, Davaadorj Duger, Bayarchimeg Batbayar

**Affiliations:** 1Department of Periodontics and Endodontics, School of Dentistry, Mongolian National University of Medical Sciences, Ulaanbaatar 14120, Mongolia; cdd25e001@gt.mnums.edu.mn (A.S.); oyunkhishig@mnums.edu.mn (O.K.); 2Department of Gastroenterology and Hepatology, School of Medicine, Mongolian National University of Medical Sciences, Ulaanbaatar 14120, Mongolia; davaadorj@mnums.edu.mn; 3Department of Endoscopy, Ulaanbaatar Songdo Hospital, Ulaanbaatar 14120, Mongolia

**Keywords:** periodontitis, ulcerative colitis, inflammatory bowel disease, case–control study

## Abstract

(1) Background: Current evidence suggests a possible association between ulcerative colitis (UC) and an increased risk of periodontitis. However, there is limited evidence regarding the severity of periodontitis in patients with UC. This study aimed to evaluate the prevalence and severity of periodontitis in patients with UC compared with healthy controls. (2) Methods: In this case–control study, 20 patients with UC and 40 age- and sex-matched controls from Ulaanbaatar Songdo Hospital underwent a full-mouth periodontal examination. Periodontitis was classified according to the 2017 classification system. Logistic regression was used to assess the relationship between UC and stage III/IV periodontitis. (3) Results: Patients with UC exhibited a significantly higher prevalence of periodontitis (95% vs. 72.5%) and stage III/IV periodontitis (55% vs. 20%, *p* = 0.011) than the controls. UC was associated with increased odds of stage III/IV periodontitis (OR = 4.0, 95% CI: 1.24–12.88, *p* = 0.02), and this association remained significant after adjusting for age and smoking status. However, after further adjustment for age and plaque levels, the association was attenuated and lost statistical significance. (4) Conclusions: Patients with UC demonstrated a higher prevalence and severity of periodontitis than healthy controls, including a significantly increased proportion of stage III/IV cases. These findings highlight the need for periodontal screening and oral hygiene education as part of the management of patients with UC in Mongolia.

## 1. Introduction

Ulcerative colitis (UC) is a chronic inflammatory disorder affecting the gastrointestinal tract. Along with Crohn’s disease (CD), UC is one of the main types of inflammatory bowel disease (IBD). UC is characterized by continuous mucosal inflammation affecting the colon and rectum and is often accompanied by erythema, erosion, and ulceration [1]. Although the precise etiology remains unclear, several factors, including genetic predisposition, environmental factors, immune response dysregulation, and gut dysbiosis, have been implicated in disease onset and progression [2]. The highest annual incidence rates of UC have been documented in Europe (24.3 per 100,000), North America (19.2 per 100,000), Asia, and the Middle East (6.3 per 100,000) [3]. In addition to intestinal involvement, IBD can manifest systemically, affecting the joints, skin, eyes, and oral cavity [4]. Oral manifestations of UC include pyostomatitis vegetans, recurrent aphthous stomatitis, atrophic glossitis, burning mouth syndrome, taste alterations, halitosis, and periodontal disease [5]. Moreover, a recent study also found a significant association between UC severity and certain oral symptoms, including dysphagia, vomiting, acidic taste, and coated tongue [6]. These manifestations may be influenced by medication use (antibiotics, corticosteroids, immunomodulators), nutritional deficiencies, oral microbiome dysbiosis, inadequate oral hygiene, and lifestyle habits such as smoking [7].

Periodontitis is an inflammatory disease that affects tooth-supporting structures and leads to progressive breakdown of the periodontal ligament and alveolar bone. It is the sixth most prevalent human disease, with severe forms affecting approximately 10–34% of the adult population, and is the leading cause of tooth loss [8,9]. In recent years, several systemic diseases, including cardiovascular diseases, diabetes mellitus, respiratory conditions, and gastrointestinal diseases, have been associated with increased periodontitis. This association is considered to be mediated by recurrent bacteremia, low-grade systemic inflammation, and autoimmune dysregulation [10,11]. Regarding gastrointestinal diseases, a recent meta-analysis reported that patients with colorectal cancer had a 20% higher risk of developing periodontitis and found a direct association between the CD, UC, and periodontitis [12]. Emerging evidence indicates that the interaction between IBD and periodontitis may occur through multiple pathways along the gum–gut axis, including hematogenous spread, enteral translocation, and immune-mediated mechanisms [13].

Several studies have reported a higher prevalence and greater severity of periodontitis in patients with IBD than in non-IBD controls [14,15,16,17,18]. Cross-sectional analyses have shown that patients with IBD tend to exhibit increased probing pocket depths and greater clinical attachment loss than healthy controls [19,20,21,22]. A recent small-sized cross-sectional study in a Norwegian UC population found a periodontitis prevalence of 74%, with 34% classified as stage III periodontitis [23].

Research on this topic is scarce in Mongolia, particularly regarding the burden of periodontitis in patients with UC. While previous studies have mainly reported clinical measurements, including probing depth and attachment loss, few have assessed disease severity using the 2017 World Workshop classification system. We hypothesize that patients with UC could have a higher prevalence and more severe periodontitis compared to controls. Therefore, this study aimed to evaluate the prevalence and severity of periodontitis in Mongolian patients with UC compared to an age- and sex-matched control group without UC.

## 2. Materials and Methods

### 2.1. Study Participants

In this case–control study, outpatients diagnosed with UC aged 18 years and above who attended the Department of Gastroenterology and Endoscopy at Ulaanbaatar Songdo Hospital from April 2024 to January 2025 were invited to participate in this study.

The sample size was calculated using the OpenEpi software, https://www.openepi.com (Boston, MA, USA) and the Fleiss method, based on an expected prevalence of periodontitis of 85% among UC individuals compared to 45% among non-UC controls. Based on an alpha error of 0.05, a power of 0.80, and a case-to-control ratio of 1:2, the required sample size was estimated to be 51 participants (17 cases and 34 controls). A total of 124 patients were diagnosed with UC at Ulaanbaatar Songdo Hospital. Patients with a history of systemic conditions that might affect periodontal tissues were excluded (i.e., patients with a history of diabetes mellitus, chronic renal disease, thyroid disease). Pregnant women, patients with fewer than six teeth, and those who had undergone periodontal treatment in the past three months were also excluded. During the study period, 68 patients attended the hospital. After applying the eligibility criteria, 26 patients were invited to participate in the study, resulting in a final case group of 20 UC patients. Forty controls with no clinical signs of UC or systemic diseases affecting the periodontal tissues, matched by age and sex to the UC group, were recruited from the same hospital. These individuals attended the outpatient clinic for routine medical examinations (Figure 1).

### 2.2. Ethical Considerations

The study was conducted in accordance with the Helsinki Declaration and approved by the Institutional Ethics Committee of the Mongolian National University of Medical Sciences (MNUMS) (Approval No: 24-25/04-02, dated 24 January 2025). All participants provided informed consent for the periodontal examination.

### 2.3. Diagnosis of UC and Data Collection

UC diagnosis was established by gastroenterologists at Ulaanbaatar Songdo Hospital, based on standard clinical assessments (tenesmus, abdominal cramps, bloody diarrhea), endoscopic findings of continuous mucosal inflammation and loss of vascular pattern, and histopathological confirmation.

Participants completed a questionnaire that collected information on sociodemographic characteristics (age, living area, and education level), smoking behavior, and oral hygiene behavior (frequency of tooth brushing and use of interdental devices). For patients with UC, the clinical data included the duration after UC diagnosis and localization (E1 = proctitis; E2 = left-sided UC; E3 = pancolitis) according to the Montreal Classification [24].

### 2.4. Periodontal Examination

All periodontal examinations were performed by a single calibrated periodontist (A.S.) at the Department of Periodontics and Endodontics, Central Dental Hospital, MNUMS. To ensure intra-examiner reliability, a separate group of 10 patients was examined twice, 48 h apart. The intra-class correlation coefficient (ICC) for repeated probing measurements was 0.85. Periodontal parameters, including full-mouth probing pocket depth (PPD), bleeding on probing (BOP), and clinical attachment level (CAL), were measured at six sites per tooth, except for the third molars. The full-mouth plaque score (FMPS) was calculated as the percentage of tooth surfaces with visible plaque, based on the plaque index (PI) recorded at six sites per tooth. The full-mouth bleeding score (FMBS) was calculated as the percentage of sites exhibiting BOP out of all probed sites. All measurements were performed using a UNC 15 (University of North Carolina) probe, and the readings were recorded to the nearest millimeter.

Participants were classified into two groups based on periodontitis severity according to the criteria of the 2017 Classification of Periodontal and Peri-Implant Diseases and Conditions [25]:Stage I/II periodontitis:
Interdental CAL of 1–4 mm;Maximum probing depth ≤ 5 mm;No tooth loss due to periodontitis.
Stage III/IV periodontitis:
Interdental CAL ≥ 5 mm;Probing depth ≥ 6 mm;Tooth mobility Grade II or higher;At least one tooth due to periodontitis (Stage III) or masticatory dysfunction, bite collapse, or less than 20 remaining teeth (Stage IV).


### 2.5. Statistical Analysis

The characteristics of the study population were expressed as means with standard deviations (SDs). Continuous variables that did not follow a normal distribution were expressed as medians with interquartile ranges (IQRs). Categorical variables were expressed as percentages with corresponding numbers. A Shapiro–Wilk test was used to assess the normality of continuous variables, including the number of teeth, FMPS, FMBS, CAL, and the number of sites with PPD ≥ 4 mm. As all continuous variables were not normally distributed, the Mann–Whitney U test was used to assess differences in continuous variables between the case and control groups. Fisher’s exact test was used to analyze categorical variables.

In the logistic regression analysis, having stage III/IV periodontitis was used as the dependent variable, with no or stage I/II periodontitis serving as the reference group. The results were presented as odds ratios (ORs) with corresponding 95% confidence intervals (CIs). After crude analysis, the data were adjusted for age, age and smoking status, and finally age and FMPS.

For all the statistical analyses, we used IBM SPSS (version 30.0; IBM, Chicago, IL, USA), and the statistical significance level was set at *p* < 0.05.

## 3. Results

### 3.1. General Characteristics

A total of 60 Mongolian individuals were recruited for this study, of which 20 were UC patients and 40 were healthy controls. The mean age of the UC patients and controls was 42.9 ± 12.14 years and 42.58 ± 12.06 years, respectively. No statistically significant differences were found between the groups regarding their sociodemographic characteristics or oral hygiene practices (*p* > 0.05), indicating baseline homogeneity between the two groups (Table 1).

### 3.2. Clinical Characteristics

The number of teeth did not differ significantly between UC patients and healthy controls. However, patients with UC had a significantly higher prevalence of periodontitis (95% vs. 72.5%; *p* < 0.05) and a greater frequency of stage III/IV periodontitis (55% vs. 20%; *p* < 0.05). CAL (4 vs. 2.5; *p* = 0.010), FMPS (71.4% vs. 54.1%; *p* = 0.002), and the number of sites with PPD *≥* 4 mm were significantly higher in UC patients (3.5 vs. 1.0; *p* < 0.001), whereas FMBS did not differ between the groups (Table 2).

Although the association between UC localization and stage III/IV periodontitis did not reach statistical significance (*p* > 0.05), a visual trend was observed, with the prevalence increasing from proctitis (E1) to pancolitis (E3) (Figure 2).

Univariate logistic regression analysis revealed that age (OR = 1.13, 95% CI: 1.06–1.20) and FMPS (OR = 1.09, 95% CI: 1.04–1.16) were significantly associated with increased odds of having stage III/IV periodontitis (*p* < 0.001 for both) (Table 3).

Finally, we tested the relationship between UC and stage III/IV periodontitis using logistic regression analysis (Table 4). In univariable logistic regression, patients with UC had significantly higher odds of having stage III/IV periodontitis compared to controls (OR = 4.0, 95% CI: 1.24–12.88). The association remained significant after adjusting for age (OR = 9.14, 95% CI: 1.72–48.65) and after adjusting for age and smoking (OR = 8.93, 95% CI: 1.60–49.81), indicating that the association was independent of age and smoking. However, after further adjustment for age and FMPS, the association did not reach statistical significance. These results suggest that the association between UC and periodontitis severity may be influenced, in part, by oral hygiene, as reflected by the FMPS.

In summary, this study found that patients with UC exhibited a significantly higher prevalence of periodontitis, compared to healthy controls (95% vs. 72.5%; *p* < 0.05). The severity of disease was also greater, with 55% of UC patients presenting with stage III/IV periodontitis (*p* = 0.011). Regression analysis indicated that UC patients had fourfold higher odds (OR = 4.0; 95% CI: 1.24–12.88) of exhibiting stage III/IV periodontitis.

## 4. Discussion

This study demonstrated significantly higher prevalence and frequency of stage III/IV periodontitis in patients with UC compared to age- and sex-matched healthy controls. These findings support our hypothesis that UC patients would exhibit greater prevalence and severity of periodontitis compared to controls. A significant relationship was observed between UC and severe periodontitis, which remained significant even after adjusting for potential confounders, such as age and smoking. However, when adjustments were made for age and the FMPS, the association was no longer significant. This suggests that poor oral hygiene may mediate the association between UC and severe periodontitis.

The present study supports the conclusions of previous meta-analyses that reported a higher prevalence of periodontitis in patients with UC [14,15]. Our findings are particularly consistent with the study by Brito et al., which reported a 90% periodontitis prevalence in UC patients, compared to 68% in the control group. Similarly, a recent investigation in a Chinese cohort found a significantly greater periodontitis prevalence among patients with UC than among healthy individuals (50.8% vs. 19.2%) [21]. However, certain studies failed to identify significant differences in periodontal disease prevalence between UC patients and controls. Grössner-Schreiber et al. reported no substantial differences in periodontal diagnosis between the groups [26]. Another recent retrospective study also found no significant difference in the frequency of periodontitis between patients with UC and healthy controls [27]. A possible explanation for these discrepancies is the use of simplified periodontal assessment tools, such as the Dutch Periodontal Screening Index. The current study used a comprehensive periodontal examination approach and, to the best of our knowledge, is one of the first to apply the 2017 Classification of Periodontal and Peri-Implant Diseases and Conditions in this context.

Our results also demonstrated that patients with UC had approximately four times greater odds of presenting with stage III/IV periodontitis than those without UC. These findings are consistent with those of previous studies that suggested a link between IBD and periodontitis [14,16,17]. One of the earliest systematic reviews reported an OR of 5.1 for periodontitis in patients with IBD [14]. A more recent case–control study involving 180 patients with IBD, including 60 with UC, observed ORs ranging from 3 to 4.5 for moderate-to-severe periodontitis. Moreover, the research indicated that the extent of inflammation in the intestinal mucosa was a significant predictor of periodontitis in both UC and CD [28]. In our study, although not statistically significant, there was a trend of increasing stage III/IV periodontitis from proctitis to pancolitis. These results suggest that greater disease extent in UC may be associated with more advanced periodontal destruction.

Although an increased prevalence of periodontitis has been reported in patients with IBD, the precise pathophysiological pathways linking these conditions remain insufficiently clarified [29]. The proposed oral–gut axis provides a biologic framework, suggesting bidirectional microbial and immune-inflammatory interactions between the intestine and the oral cavity. Regarding microbial factors, a recent review highlights the role of *Klebsiella*, *Porphyromonas gingivalis*, and *Fusobacterium nucleatum*, as well as the pro-inflammatory effects of altered microbial metabolites, including short-chain fatty acids. These microorganisms, enriched in the oral cavity of IBD patients, are also strongly associated with the development of periodontitis. Therefore, studies suggest that oral ecological dysregulation in IBD patients may cause the development of periodontitis [30]. In parallel, elevated concentrations of proinflammatory cytokines, such as interleukin (IL)-1β, IL-6, tumor necrosis factor (TNF)-α, and matrix metalloproteinase (MMP)-8 have been reported in the saliva and gingival crevicular fluid of patients with UC and CD [13]. Conversely, experimental data indicate that periodontitis may aggravate UC by inducing intestinal dysbiosis and promoting pro-inflammatory cytokine secretion, including TNF-α and IL-6 in mice. These alterations might lead to a compromised intestinal barrier, which may worsen UC severity [31]. Furthermore, oral *Fusobacterium nucleatum* has been implicated in UC pathogenesis by translocating to the gut and enhancing intestinal permeability [32]. Beyond UC, periodontal pathogens have been implicated in the pathogenesis of various systemic diseases through the oral–gut axis, including colorectal cancer, rheumatoid arthritis, diabetes mellitus, Alzheimer’s disease, and atherosclerotic cardiovascular disease [33]. Despite these insights, current evidence remains limited, and further mechanistic studies are warranted to establish pathways underlying the bidirectional association between UC and periodontitis.

Notably, this study highlighted the role of plaque accumulation in the association between UC and periodontitis severity. After adjusting for age and the FMPS, the association between UC and stage III/IV periodontitis was not significant, emphasizing the potential importance of plaque control in the progression of periodontal disease. Consistent with this, Brito et al. found that patients with IBD harbored higher levels of subgingival bacterial load in untreated sites than controls [34]. Similarly, Baima et al. reported that the association between IBD and periodontitis was particularly pronounced among patients aged 36–50 and 51–65 years [28]. Taken together, these findings emphasize the need to account for confounding factors when evaluating the relationship between UC and periodontitis.

To our knowledge, this is the first clinical study to report a link between UC and severe periodontitis in the Mongolian population. In addition, several strategies were employed to minimize confounding, including the application of strict inclusion and exclusion criteria, the use of a matching method, and the implementation of multivariate statistical analysis. Nonetheless, this study has limitations, including a small sample size and a single-center design, which may restrict the generalizability of the results. Considering the multifactorial pathogenesis of both conditions, residual confounders cannot be ruled out. Therefore, future research should include larger, multicenter, longitudinal studies to confirm these findings. Particular attention should be given to the role of disease extent and activity, as well as the effects of UC medications. Furthermore, investigations into pro-inflammatory mediators such as IL-1β, IL-6, and TNF-α, together with detailed microbiome profiling, may provide clearer insights into the biological mechanisms underlying this association.

## 5. Conclusions

The present study demonstrated a higher prevalence and increased proportion of stage III/IV periodontitis in Mongolian patients with UC than in healthy controls. The association between UC and severe periodontitis was independent of age and smoking status. However, the attenuation of this association after adjusting for plaque levels suggests that poor oral hygiene may have a mediating role in this relationship. These findings emphasize the need for periodontal screening and oral hygiene education in the management of patients with UC in Mongolia. In addition, future studies should incorporate pro-inflammatory mediator analyses in order to clarify the mechanistic association and employ longitudinal designs to explain causality between UC and periodontal disease.

## Figures and Tables

**Figure 1 ijerph-22-01355-f001:**
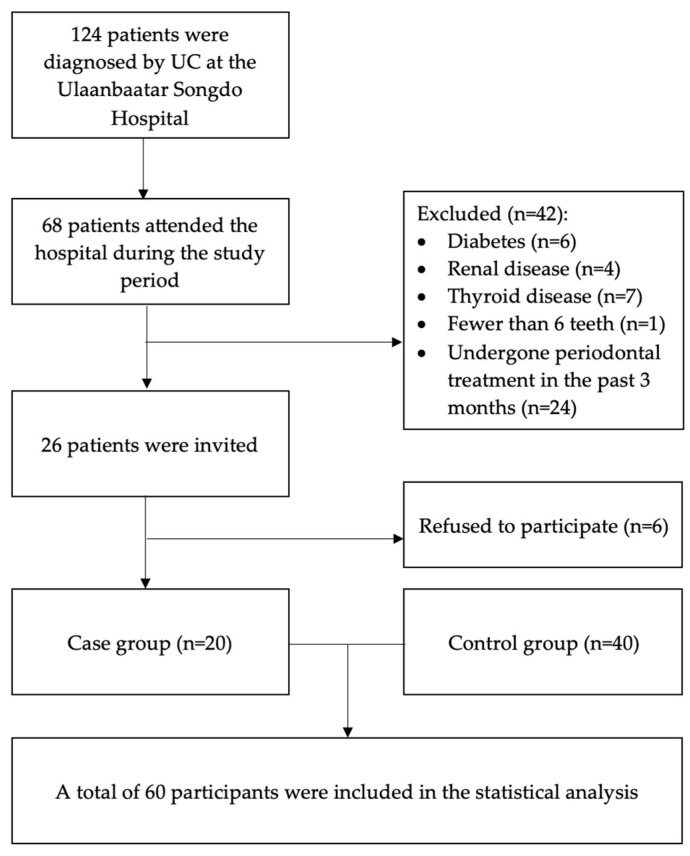
Flowchart of study participants.

**Figure 2 ijerph-22-01355-f002:**
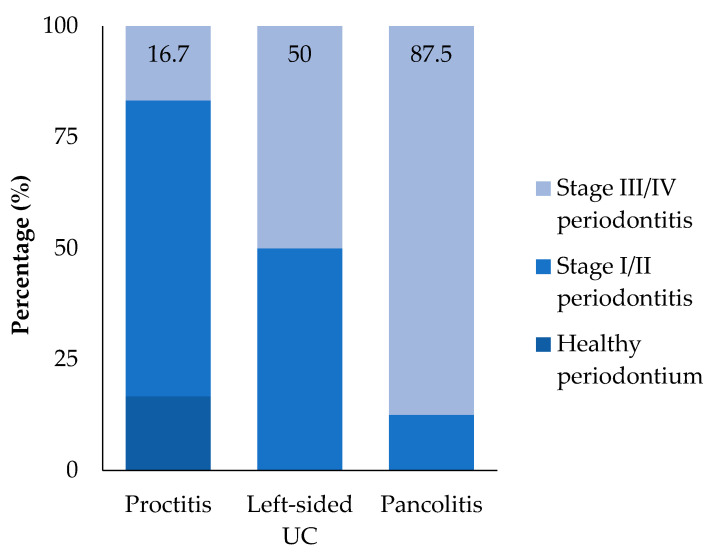
Percentage of UC patients with stage III/IV periodontitis according to UC localization.

**Table 1 ijerph-22-01355-t001:** Characteristics of the study population and the disease.

Variables	UC (*n* = 20)	Controls (*n* = 40)	*p*-Value
Age (years)	42.9 ± 12.14	42.58 ± 12.06	0.875
Gender: male, % (n)	40 (8)	40 (16)	1.00
Area, % (n) Urban Rural	80 (16) 20 (4)	90 (36) 10 (4)	0.422
Education level, % (n) Low Intermediate High	- 15 (3) 85 (17)	32.5 (13) 67.5 (27)	0.218
Years with UC diagnosis	1.79 (2.68)	-	
Localization, % (n) E1 E2 E3	30 (6) 30 (6) 40 (8)	-	
Oral health practices, % (n) Flossing	45 (9)	55 (22)	0.586
Brushing frequency, % (n) ≤Once/day Twice/day ≥Three times/day	45 (9) 55 (11) -	27.5 (11) 72.5 (29) -	0.246
Smoking, %(n) Yes No	25 (5) 75 (15)	85 (34) 15 (6)	0.481

Data are presented as mean ± SD, and percentages (number). UC, ulcerative colitis. The hyphen indicates no participants in this category.

**Table 2 ijerph-22-01355-t002:** Comparison of periodontal parameters between UC patients and controls.

Variables	UC (*n* = 20)	Controls (*n* = 40)	*p*-Value
Diagnosis, % (n)			0.011
Healthy periodontium	5 (1)	27.5 (11)	
Stage I/II periodontitis	40 (8)	52.5 (21)	
Stage III/IV periodontitis	55 (11)	20 (8)	
Periodontal parameters,			
median (IQR)			
Number of teeth	24 (5)	25 (5)	0.482
FMPS	71.4 (24.62)	54.1 (21.85)	0.002
FMBS	27.68 (15.73)	23.7 (17.79)	0.433
CAL	4 (2)	2.5 (4)	0.010
Number of PPD ≥ 4 mm	3.5 (5)	1 (3)	<0.001

Data are presented as percentages (numbers). The *p*-values correspond to the Mann–Whitney U test for continuous variables and Fisher’s exact test for categorical variables. FMPS: full-mouth plaque score; FMBS: full-mouth bleeding score; CAL: clinical attachment loss.

**Table 3 ijerph-22-01355-t003:** Logistic regression analysis for association with increased risk of stage III/IV periodontitis.

Variables	OR (95% CI)	*p*-Value
Age (years)	1.13 (1.06–1.20)	<0.001
Sex: Male	0.47 (0.14–1.54)	0.211
Smoking	1.43 (0.36–5.66)	0.611
FMPS (%)	1.09 (1.04–1.16)	<0.001
FMBS (%)	1.03 (0.99–1.07)	0.106

Data are presented as odds ratios (ORs) with 95% confidence intervals (95% CI). OR: odds ratio; CI: confidence interval; FMPS: full-mouth plaque score; FMBS: full-mouth bleeding score.

**Table 4 ijerph-22-01355-t004:** Association between UC and stage III/IV periodontitis.

UC	Stage III/IV Periodontitis	
	OR (95% CI)	*p*-Value
Crude	4.0 (1.24–12.88)	0.020
Adjusted for age	9.14 (1.72–48.65)	0.010
Adjusted for age and smoking	8.93 (1.60–49.81)	0.012
Adjusted for age and FMPS	4.86 (0.77; 30.50)	0.092

Data are presented as odds ratios (ORs) with 95% confidence intervals (95% CI). OR: odds ratio; CI: confidence interval; FMPS: full-mouth plaque score.

## Data Availability

The data underlying this study are available from the corresponding author upon request.

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
