# Peer review of "Prevalence and Severity of Periodontitis in Patients with Ulcerative Colitis: A Case–Control Study"

_ijerph, 2025, doi:10.3390/ijerph22091355_

Round 1

Reviewer 1 Report

Comments and Suggestions for Authors

I would like to congratulate the authors for their research titled "Increased Prevalence and Severity of Periodontitis in Patients with Ulcerative Colitis: A Case-Control Study." Considering the relationship between periodontal diseases and systemic conditions, especially gastrointestinal diseases, the scientific literature indeed needs more research and publications in this area. However, after reviewing your manuscript, I believe that it is not publishable in its current form. A number of revisions are necessary, and the manuscript should be thoroughly reviewed again following these corrections.

  • Although the Introduction section provides adequate information about ulcerative colitis and periodontitis, it lacks sufficient discussion on the relationship between systemic diseases—particularly gastrointestinal disorders—and periodontal diseases. Your study's null hypothesis/hypotheses should be clearly stated in the final paragraph of the Introduction, and the acceptance or rejection of these hypotheses should be explicitly mentioned in the first paragraph of the Discussion.

  • Ethical considerations regarding your study (name of the ethics committee, approval number, compliance with the Declaration of Helsinki, informed consent procedures, etc.) must be presented in a separate subheading within the Methods section.

  • Who diagnosed ulcerative colitis? Was intra- or inter-examiner calibration performed? If so, these procedures should be described in detail. The same applies to the periodontal examination. Calibration is a crucial aspect and must be addressed comprehensively. Was a pilot study or assessment conducted? If so, what were the statistical findings? If not, this constitutes a significant limitation.

  • The diagnostic process for ulcerative colitis should be described in more detail. How is ulcerative colitis diagnosed? What are the symptoms? What tests are applied? Are there serological, microbiological, or blood tests involved? This should be elaborated under a separate subheading dedicated to the diagnosis of ulcerative colitis.

  • A sample size calculation section should be included.

  • Although the statistical analyses are commendable, for instance, it is understood that age and FMPS significantly increase the risk of periodontitis. However, what is the combined effect of being older and having a high FMPS on periodontitis? Similarly, how does smoking influence the condition in a UC patient compared to a non-smoker? Advanced statistical analyses should be conducted and presented to explore these interactions.

  • Additionally, please include intraoral photographs and radiographs of one individual with UC and one without UC.

  • While the Discussion section is adequate in the context of UC, as mentioned regarding the Introduction, there are numerous studies examining the association between other gastrointestinal (and systemic) diseases and periodontitis. This body of literature should be incorporated to substantially strengthen your Discussion.

  • Moreover, beyond systemic diseases, you should discuss the relationship between the other variables evaluated in your study and periodontitis, supported by relevant literature.

  • What are your recommendations for future research? Although you have touched on the strengths and limitations of your study, these aspects should be elaborated further.

I look forward to reviewing your revised manuscript. Best regards and good luck with your research.

Comments on the Quality of English Language

The English could be improved to more clearly express the research.

Author Response

Thank you for your valuable comments. 

Reviewer 2 Report

Comments and Suggestions for Authors

This manuscript investigates a clinically relevant and timely research question—the potential association between ulcerative colitis (UC) and periodontitis. The methodology of the study is appropriately designed, with clearly defined aims, a relevant case-control approach, and transparent procedural details.

The primary limitation of the manuscript lies in its small sample size and overall brevity. To strengthen its scientific merit, the authors should enrich the introduction and discussion with a more comprehensive review of relevant literature and deeper contextual analysis.

Additional comments are outlined below:

  1. The statement 'Ulcerative colitis (UC) is associated with an increased risk of developing periodontitis' should be rephrased in the abstract. As current evidence only suggests a possible association, it cannot be presented as an established fact.

  1. Please expand the introduction by elaborating further on ulcerative colitis, with a focus on its clinical manifestations in the oral cavity. Additionally, incorporate references to relevant studies and current literature to provide a comprehensive overview of the topic.

  1. Who constituted the control group? Could you clarify the characteristics of these patients, considering they were recruited from the same hospital as the experimental group?

  1. What criteria and statistical methods were employed to determine the adequacy of the sample size of 20 patients? Please provide details on the sample size calculation to justify its statistical power and relevance for the study objectives.

  1. The methods used to assess FMPS and FMBS should be clearly described, or an appropriate reference should be cited to clarify how these indices were determined.

  1. The discussion could be strengthened by a comprehensive analysis of the underlying biological mechanisms potentially connecting ulcerative colitis to periodontal tissue destruction, supported by appropriate references. Please provide an expanded discussion consisting of at least ten sentences elaborating on this topic.
  2. The conclusion should explicitly outline precise recommendations for future research, encompassing the evaluation of proinflammatory mediator analyses and the adoption of diverse methodological approaches or study designs.

Author Response

Thank you for your insightful comments. 

Reviewer 3 Report

Comments and Suggestions for Authors

Dear Author, 

Please find my comments attached 

Regards

Author Response

Thank you for your important comments. 

Reviewer 4 Report

Comments and Suggestions for Authors

This manuscript is an epidemiological study examining the relationship between ulcerative colitis and severity of periodontal disease. I think that this manuscript is highly valuable. However, I think there are several improvements that could be made to this manuscript to make it more valuable.

  1. Lines 76-78: To avoid sampling bias, please provide more detail information on how you selected 20 patients from the 124 UC patients.
  2. Discussion section: This study only compares the relationship between US and the severity of periodontal disease with other epidemiological studies. Please describe the possible mechanisms by which periodontal inflammation affects US with reference to other manuscripts. (ex: BMC Oral Health. 25(1):894, 2025; Front Cell Infect Microbol. 15:1564169, 2025)

Author Response

Thank you for your valuable comments. 

Reviewer 5 Report

Comments and Suggestions for Authors

Thank you for the opportunity to review the paper, “Increased Prevalence and Severity of Periodontitis in Patients with Ulcerative Colitis: A Case-Control Study.” The manuscript is well structured. However, the following aspects would benefit from further revision.

1. While the introduction section clearly presents the need for the study, the study's insight would be enhanced by adding a theoretically based hypothesis that UC patients have a higher prevalence of periodontitis.

2. In the methods section, presenting the participant recruitment process as a flowchart would improve readability. Please add the rationale for determining the appropriate sample size for this case-control study.

Please include the statistical software used for the analysis.

3. In the results section, Table 1 presents the results of the homogeneity test between participants. We recommend adding that the general characteristics of the two groups were homogeneous.
Please display p-values consistently to three decimal places. Currently, the second and third digits are written interchangeably.

Author Response

Thank you for your valuable comments. 

Round 2

Reviewer 1 Report

Comments and Suggestions for Authors

I congratulate the authors on their successful research and the resulting manuscript. I have carefully reviewed this version as I did with the previous one. I appreciate that the gaps I pointed out—whether methodological, statistical, or in terms of literature and background—have been thoroughly addressed and revised. In its current form, the manuscript is acceptable for publication.

Reviewer 2 Report

Comments and Suggestions for Authors

The authors have adequately addressed all the reviewers' comments.

Reviewer 4 Report

Comments and Suggestions for Authors

Thank you for revising the manuscript according to my comments.

There is no additional comment regarding this revised manuscript.